# Index of Cancer-Associated Fibroblasts Is Superior to the Epithelial–Mesenchymal Transition Score in Prognosis Prediction

**DOI:** 10.3390/cancers12071718

**Published:** 2020-06-28

**Authors:** Ying-Chieh Ko, Ting-Yu Lai, Shu-Ching Hsu, Fu-Hui Wang, Sheng-Yao Su, Yu-Lian Chen, Min-Lung Tsai, Chung-Chun Wu, Jenn-Ren Hsiao, Jang-Yang Chang, Yi-Mi Wu, Dan R. Robinson, Chung-Yen Lin, Su-Fang Lin

**Affiliations:** 1National Institute of Cancer Research, National Health Research Institutes, Miaoli County 35053, Taiwan; ingridko@nhri.edu.tw (Y.-C.K.); cyl@nhri.edu.tw (Y.-L.C.); 090107@nhri.edu.tw (M.-L.T.); chungcwu@nhri.edu.tw (C.-C.W.); em50010@email.ncku.edu.tw (J.-Y.C.); 2Institute of Bioinformatics and Structural Biology, National Tsing-Hua University, Hsinchu 30013, Taiwan; s108080806@m108.nthu.edu.tw; 3National Institute of Infectious Diseases and Vaccinology, National Health Research Institutes, Miaoli County 35053, Taiwan; mschsu@nhri.edu.tw (S.-C.H.); fhwang0919@gmail.com (F.-H.W.); 4Graduate Institute of Medicine, College of Medicine, Kaohsiung Medical University, Kaohsiung City 80708, Taiwan; 5PhD Program in Tissue Engineering and Regenerative Medicine, National Chung Hsing University, Taichung City 40227, Taiwan; 6Institute of Information Science, Academia Sinica, Taipei 11529, Taiwan; daniel0523@gmail.com (S.-Y.S.); cylin@sinica.edu.tw (C.-Y.L.); 7Translational Cell Therapy Center, Department of Medical Research, China Medical University Hospital, Taichung City 40447, Taiwan; 8Department of Otolaryngology, Head and Neck Collaborative Oncology Group, National Cheng Kung University Hospital, College of Medicine, National Cheng Kung University, Tainan 70403, Taiwan; hsiaojr@mail.ncku.edu.tw; 9Department of Internal Medicine, National Cheng Kung University Hospital, College of Medicine, National Cheng Kung University, Tainan 70403, Taiwan; 10Michigan Center for Translational Pathology, University of Michigan, Ann Arbor, MI 48109, USA; yimiwu@med.umich.edu (Y.-M.W.); danrobi@med.umich.edu (D.R.R.); 11Department of Pathology, University of Michigan, Ann Arbor, MI 48109, USA

**Keywords:** cancer-associated fibroblasts, epithelial–mesenchymal transition, oral cancer, tumor stroma, prognosis prediction

## Abstract

In many solid tumors, tissue of the mesenchymal subtype is frequently associated with epithelial–mesenchymal transition (EMT), strong stromal infiltration, and poor prognosis. Emerging evidence from tumor ecosystem studies has revealed that the two main components of tumor stroma, namely, infiltrated immune cells and cancer-associated fibroblasts (CAFs), also express certain typical EMT genes and are not distinguishable from intrinsic tumor EMT, where bulk tissue is concerned. Transcriptomic analysis of xenograft tissues provides a unique advantage in dissecting genes of tumor (human) or stroma (murine) origins. By transcriptomic analysis of xenograft tissues, we found that oral squamous cell carcinoma (OSCC) tumor cells with a high EMT score, the computed mesenchymal likelihood based on the expression signature of canonical EMT markers, are associated with elevated stromal contents featured with fibronectin 1 (Fn1) and transforming growth factor-β (Tgfβ) axis gene expression. In conjugation with meta-analysis of these genes in clinical OSCC datasets, we further extracted a four-gene index, comprising FN1, TGFB2, TGFBR2, and TGFBI, as an indicator of CAF abundance. The CAF index is more powerful than the EMT score in predicting survival outcomes, not only for oral cancer but also for the cancer genome atlas (TCGA) pan-cancer cohort comprising 9356 patients from 32 cancer subtypes. Collectively, our results suggest that a further distinction and integration of the EMT score with the CAF index will enhance prognosis prediction, thus paving the way for curative medicine in clinical oncology.

## 1. Introduction

Head and neck cancer as a whole is heterogeneous. Gene expression-based clustering methods have established four molecular subgroups of head and neck cancer tissues: basal (BA), mesenchymal (MS), classical (CL), and atypical (AT) [1,2]. By inclusion of more samples from multiple cohorts, Keck et al. [3] recently identified an inflammatory mesenchymal (IMS) subtype (33% of 938 samples analyzed) that displayed prominent immune infiltration, a mesenchymal phenotype, and favorable clinical outcomes. Paradoxically, in cancers of the breast, lung, and digestive tracts, the mesenchymal subtype is associated with poor prognosis [4,5,6,7].

Tumor cells with the mesenchymal phenotype are thought to derive from undergoing an epithelial–mesenchymal transition (EMT) [8]. EMT is associated with cancer stem cell characteristics, including a more migratory and invasive phenotype that opts for metastasis and drug resistance [6,9]. In vivo, despite the lack of a consensus molecular program underlying EMT, emerging evidence indicates that EMT is context dependent, epigenetic driven, and is likely to be metastable [10,11]. In vitro, EMT gene signatures can be variously derived by treating cells of interest with diverse stimuli, or ectopic expression of master regulatory genes for EMT. For example, using Twist or Snail-inducible non-small cell lung cancer cell models, Salt et al. [12] identified a 14-gene signature, including 9 mesenchymal and 5 epithelial markers, associated with growth survival in response to nutrition deprivation. With these regards, thanks to the high-resolution single-cell RNA-seq methodology, Puram et al. [13] recently revealed the ecosystems of ~6000 cells from 18 treatment-naïve oral cavity squamous cell carcinoma (OSCC) specimens, including 5 matched lymph node metastases. By deconvolution of bulk expression data, the authors proposed a partial-EMT program, notably presented in a redefined malignant-basal subtype consisting of >70% of oral cavity tumors in the cancer genome atlas (TCGA) cohort. Similar to EMT, the authors provided evidence that the tumor microenvironment (TME) surrounding the leading edges of OSCC tissues with partial EMT plays critical roles in metastasis.

The tumor microenvironment (TME) is regarded as the non-malignant part of a tumor that harbors infiltrated immune cells, vascular/lymphatic endothelial cells, and fibroblasts, also known as cancer-associated fibroblasts (CAFs) [14]. Despite their significant importance, a consensus statement of CAFs was not formulated until recently [15]. One of the well-established TME activating signals for fibroblasts is the transforming growth factor-β (TGFβ) superfamily. Secreted by various types of cells, these multifunctional peptides drive differentiation of local fibroblasts into CAFs. The TGFβ axis is an evolutionarily conserved molecular pathway that exerts anti- and pro-tumor activities during the initial and late stages of tumorigenesis, respectively. The comparative genomic loci of most human and mouse TGFβ axis genes, including TGFB1–3, TGFBR1–3, and TGFBI, were mapped ~20 years ago [16], which paved the way for numerous functional studies of individual TGFβ genes in tumor biology. Importantly, this pathway comprises multiple targets attributable to therapeutic resistance [17,18], linking its importance to clinics.

Another collection of essential players residing in the TME is the extracellular matrix (ECM). Composed of structurally and functionally diverse glycoproteins and proteoglycans, the web-like ECM is topologically and spatiotemporally dynamic [19]. ECM dynamics in the TME can result from rapid changes in ECM composition (e.g., fibronectin, collagen, laminin), elevated activities of matrix metalloproteinases (e.g., matrix metallopeptidase (MMP), a disintegrin and metalloproteinase with thrombospondin type 1 (ADAMTS)), or increased levels of cross-linking enzymes (e.g., lysyl oxidase (LOX), procollagen-lysine,2-oxoglutarate 5-dioxygenase (PLOD)). In addition, due to its highly charged polysaccharide modifications, the ECM traps the pro- and mature forms of growth factors and cytokines, serving as a reservoir to limit the accessibility of ligands to their cognate receptors in the normal steady-state condition. Given these highly dedicated functions orchestrated by ECM, therapeutic interventions for TME ECM are emerging [20,21,22].

Previously, in the course of analyzing an RNA-seq dataset comprising five Taiwanese OSCC and two immortalized keratinocyte cell lines (GSE150469), we noticed that compared to the median values, OSCC OC3 overexpressed *KRT5*, *KRT14*, *SNAI2*, *TWIST1*, *VIM*, *ZEB1*, and *ZEB2* (2–22×) and under-expressed *CDH1*, *EPCAM*, *KRT8*, *SNAI1*, and *TWIST2* (0.01–0.15×), suggesting its basal-like propensity. By contrast, the other four OSCC cell lines maintained moderate levels of *CDH1* and *EPCAM*, two surface markers indicative of epithelial states. In addition, our prior results showed that one of the four epithelial-like OSCC, TW2.6, displayed prominent cell cohesiveness in vitro and partial EMT-like invasion in vivo [23]. Here, in order to better understand the relationship between EMT and stromal infiltration in oral cancer, OC3 and TW2.6 were selected for xenograft studies. We anticipated that the gene expression profiles of the host/murine and graft/human can serve as separable blueprints for the stroma and tumor, respectively.

## 2. Results

### 2.1. The Oral Cancer Cell Line with a Higher EMT Score Is Associated with Stronger Stromal Content In Vivo

Salt et al. [12] proposed a 14-gene matrix, dubbed the EMT score, to evaluate the likelihood of an epithelial cell transiting to a mesenchymal state. OC3 and TW2.6 are betel-quid-associated OSCC cells, with inherent high and low EMT scores, respectively (Figure 1a). Of note, OC3 and TW2.6 displayed similar levels of clonogenicity, spheroid formation, and proliferation rate in vitro (Appendix A). As a step to assess EMT-associated stromal infiltration in vivo, immunodeficient NOG mice were subcutaneously implanted with OC3 and TW2.6 cells, respectively. In two independent experiments, NOG mice inoculated with OC3 cells repeatedly exhibited a significant delay in tumor formation compared to animals injected with TW2.6 cells (Figure 1b). In addition, more than 50% of mice in the OC3 group displayed swollen spleens compared to the control mice upon euthanasia, whereas the TW2.6 group had the opposite phenomenon (Appendix A). As the spleen is the principal organ manufacturing immune components, these results implied that OC3 might be strongly immunogenic to the NOG host, yet TW2.6 is an immune-evasive subtype.

The total RNA of six OC3 and five TW2.6-derived xenografts was subjected to mRNA-seq analysis. The bioinformatic pipelines *Xenome* [24] and *XenofilteR* [25] were independently applied to assign the tumor and stromal reads in each sample (Figure 1c). Evidently, both the murine read counts and stromal fractions of the OC3 xenograft tissues are higher than that of the TW2.6 group (32.5% vs. 12.7%, *p* < 0.001) (Figure 1c and Appendix A). Taken together, the high EMT score OC3 cell line was likely immunogenic, retarded in tumor growth, and associated with strong stromal infiltration in vivo, suggesting an indispensable role of the host stroma for OC3 survival.

### 2.2. OC3 Tumor Cells Displayed a Strong Innate Immune Response in Xenograft Tissues

As a first step to explore what molecular pathways are enriched in the OC3 and TW2.6 xenograft tissues, *GSEA* [26] and *limma* [27] were employed to identify differentially expressed genes (DEGs) and biological pathways in the human reads, representing the OC3 and TW2.6 tumor intrinsic factors. Differential expression analysis by using the *GSEA Hallmark* database revealed that various immune response pathways (false discovery rate (FDR) q-val = 0.000) and MYC/E2F targets (FDR q-val ≤ 0.015) are enriched in the OC3 tumors and TW2.6 tumors, respectively (Appendix A). Furthermore, DEGs identified by *limma* (fold change (FC) > 1.5) were subjected to *DAVID* portal [28] for gene ontology (GO) analysis (Figure 2a). In comparison, the results of *DAVID* and *GSEA* analyses agree with each other, in that both bioinformatic approaches showed statistical enrichments of immune response pathways in OC3 tumors and transcription/translation pathways in TW2.6 tumors, respectively.

Specifically, the upregulated DEGs in OC3 tumors are enriched in the type I interferon signaling pathway, interferon alpha/beta/gamma signaling pathway, antigen processing and presentation pathway, and innate immune responses. As a further support for the pathway enrichment analysis, the expression levels of distinct immune-related and proliferation-related genes are plotted in Figure 2b. OC3 tumors show elevated expression of antigen processing and presentation markers, beta-2-microglobulin (B2M), and polymorphic HLA-encoded alpha chain (HLA-A). Interferon signaling pathway-related genes, IFIT1, IFI44, and IFNGR1, were also overexpressed in OC3 tumors. Apoptosis-related DEGs, including CASP8 and TNFSF10, were overexpressed in OC3 tumor cells, which might explain the tumor growth retardation of OC3 xenografts (Appendix A). As STAT1 is a crucial hub gene responsible for various cellular immune responses [29] and OSCC pathogenesis [30], immunohistochemical staining of signal transducer and activator of transcription 1 (STAT1) was performed to validate the RNA-seq results (Figure 2c). Clearly, results from both assays agreed with each other in that human STAT1 was strongly detected in the tumor part of OC3 tissue sections, whereas mouse STAT1 was more evident in the stroma parts of TW2.6 tissue sections. On the other hand, TW2.6 tumors exhibited increased expression of proliferation-related genes, including MYC, ribosomal proteins (RPL37, RPS18, and RPS5), and mitochondrial ribosomal protein (MRPL36). These overexpressed genes supported the rapid growth phenotype of TW2.6 tumors observed in vivo (Figure 1b).

We also plotted the expression of the 14 genes that constitute the EMT score (Figure 1a) in each xenograft tissue (Appendix A). Not all mesenchymal markers of OC3 cells were retained in vivo, e.g., expression levels of ZEB2 and TWIST1 are similar in both groups. Similarly, only three out of five epithelial markers, namely, CLDN4, CLDN7, and CDH1, were preferentially expressed in the TW2.6 group, suggesting that EMT markers identified in vitro should be interpreted with caution.

### 2.3. Upregulated Expression of Extracellular Matrix (ECM) and Fibroblast-Related Genes in OC3 CDXs

Next, the same bioinformatics procedures were applied to analyze the murine reads. To avoid ambiguity, the murine/stroma part will be noted with lowercases hereafter. As shown in Appendix A and Figure 3a, pathways enriched in the OC3 and TW2.6 stroma were apparently different from their tumor counterparts. The OC3 stroma showed fibronectin (Fn1)—and transforming growth factor beta (TGFβ)—related GO terms, including glycosaminoglycan binding, cell adhesion, heparin binding, osteoblast differentiation, extracellular matrix organization, molecules associated with elastic fibers, embryonic cranial skeleton morphogenesis, ECM–receptor interactions, pathways in cancer, and positive regulation of mesenchymal cell proliferation. In addition, genes of the wingless-type MMTV integration site family (Wnt) signaling pathway (e.g., Wnt9A) and metalloproteinases (e.g., Mmp14) were also detected (Figure 3b). Together, these data strongly suggest that the oc3-stroma actively underwent tissue remodeling. It is worth noting that in the TW2.6 strom-enriched GO terms include host keratinocyte differentiation, epidermis development, innate immune responses, and major histocompatibility complex (MHC)-I antigen presentation (Figure 3a,b). Furthermore, the expression levels of macrophage markers (Adgre1, Sirpa), dendritic cell marker (Itgax), and the host ‘don’t eat me’ signal Cd47 [31] are similar between the OC3 and TW2.6 stroma.

Since Fn1 appeared in various GO terms enriched in the OC3 stroma, we further validated this bioinformatic finding by using dual immunohistochemical staining. Specifically, tissue sections of six OC3 and six TW2.6 xenografts were co-stained for human KRT18 and murine Fn1. As depicted in Figure 3c and Appendix A, stronger Fn1 staining is repeatedly detected in the OC3 stroma compared to that of the TW2.6 stroma. Quantitative analysis of Fn1 staining revealed that the difference is statistically significant (50.8% ± 2.5 vs. 18.8% ± 1.5, *p* < 0.0001, Figure 3c). Independently, by using Masson’s trichrome staining, OC3 stroma also harbored a higher extent of collagenous connective tissue fibers than the TW2.6 stroma did (30.5% ± 1.3 vs. 16.5% ± 2.1, *p* < 0.0001, Appendix A), reinforcing the notion that the OC3 stroma was rich in ECM.

### 2.4. The Clinical Relevance of Fn1 and TGFβ Axis Genes in Clinical OSCC Datasets

We were aware that several obvious caveats are inherent to the NOG xenograft study, including the fact that the lack of complete immune systems and human orthologous genes might not be entirely functional in mice. Through the literature search, we chose the TGFβ signaling pathway for further in-depth study since it is evolutionarily conserved in humans and mice [16], and has been repeatedly documented in head and cancer pathogenesis [18,32,33,34,35]. First, we consulted the single-cell RNA-seq dataset GSE103322, comprising ~6000 malignant and non-malignant cells from 18 OSCC tissues, for the cell types that express genes of our interest. Specifically, we extracted expression levels of FN1, TGFβ axis genes (ligands, receptors, and TGFBI), and activated fibroblast markers (FAP, ACTA2/αSMA, CSF1R) from our xenograft datasets (Figure 4a and Appendix A) and GSE103322 (Figure 4b), respectively. When compared side by side, the enriched expression of oc3-Fn1, Tgfb3, Tgfbr2, and Tgfbr3 recapitulated the human orthologs in the stromal compartment of GSE103322. It is tempting to envisage that fibroblasts residing in the OC3 stroma, including prototypical and endothelial derived [36], are likely to be the main source that expressed these genes. Of note, conventional activated fibroblast markers were expressed similarly in the OC3 and TW2.6 stroma (Appendix A), suggesting that activated fibroblasts might also exist in the TW2.6 stroma. Next, we observed high yet not significantly different levels of Tgfbi (Figure 4a) and TGFBI (Appendix A) were detected in the OC3 and TW2.6 xenografts, which is consistent with comparable levels of TGFBI detected in the stroma and tumor parts of clinical samples (Figure 4b). Finally, despite TGFB2 and TGFBR3 being preferentially expressed in the OC3 tumor group (Appendix A), in clinical samples, both genes were not significantly enriched in the tumor cell type (Figure 4b). We proposed a further separation of tumor cells with malignant-basal (*n* = 1148) from atypical and classical subtypes in the mixed tumor cell population (*n* = 2539) [13] might help to resolve this inconsistency.

### 2.5. The Index of Cancer-Associated Fibroblasts Is Better than the EMT Score in Overall Survival Prediction for Cancer Patients

To further explore the associations of FN1 and TGFβ axis genes with clinical outcomes, the 40 OSCC tissues in the dataset GSE37991 were used. This is a microarray-based transcriptome profile of 40 matched pairs of betel quid-associated OSCC and adjacent normal tissues, dubbed NCKU-OrCA-40TN, established in our institute previously [37]. Prior Cox proportional hazards model assessments revealed that the EMT score, but not lymph node metastasis or angiolymphatic invasion, is a statistical risk factor for overall survival of the 40 OSCC patients (HR = 4.2; 95% CI, 1.3–14; *p* = 0.015) (Appendix A). The same analysis was performed to evaluate the expression of FN1 and each TGFβ axis gene in bulk tissues. The results showed that TGFB2 (HR = 4.9; 95% CI, 1.4–17; *p* = 0.015) and TGFBI (HR = 11; 95% CI, 2.4–50, *p* = 0.0021) also show statistical significance (Appendix Ac). Collectively, these meta-analyses suggest that in a given tissue, aggregated expression of FN1, TGFBR2, TGFB2, and TGFBI might serve as an index for the extent of cancer-associated fibroblasts (CAFs). To test whether the CAF index is similar to the EMT score, an adverse prognosis marker, both the EMT score and the CAF index, of each tumor tissue in GSE37991 were computed, followed by dichotomized Kaplan–Meier survival analysis and Cox proportional hazards model assessment (Figure 5). As expected, both the EMT score and CAF index are powerful predictors, with the latter gaining greater log-rank statistics (6.87 vs. 16.6) and better assignments for patients with shorter survival (red dots in 5a and 5b). The index of CAFs also outperformed the EMT score in multivariate Cox model analysis, i.e., CAF index (HR = 12.5; 95% CI, 2.08–74.9, *p* = 0.006) vs. EMT score (HR = 1.0; 95% CI, 0.24–4.2, *p* = 0.992) (Figure 5c). This result validated that the CAF index reversely correlated with overall survival and is a better predictor than the EMT score for the 40 OSCC patients.

To assess whether the CAF index can serve as a surrogate for poor prognosis in other cancer types, the EMT score and CAF index of each sample in the TCGA pan-cancer cohort (*n* = 9356) comprising 32 cancer subtypes were computed followed by dichotomized Kaplan–Meier survival analysis and Cox model assessment as described above (Figure 6a). Notably, the CAF index is also superior to the EMT score in the Kaplan–Meier plots (log-rank statistics 109.4 vs. 29.89) and in multivariate Cox model analysis (CAF index (HR = 1.5; 95% CI, 1.3–1.6, *p* < 0.001) vs. EMT score (HR = 1.1; 95% CI, 1.0–1.2, *p* = 0.014)), indicating that the applicability of the CAF index is not only limited to OSCC patients. Furthermore, meta-analysis of the CAF index with MET500 [38], a dataset comprising integrative clinical genomics of 811 metastatic tumor samples, revealed a negative correlation between the CAF index and the tumor DNA content (r = −0.383, *p* < 2.2 × 10^−16^) (Figure 6b, left), suggesting that a bulk met tissue is also a composite of tumor cells and stroma. In addition, among the 22 primary sites in MET500, oral cancer has the second highest CAF index, just next to the liver (Figure 6b, right).

### 2.6. Comparison of CAF Index and EMT Score Composed of 76-Gene Signature (76GS)

Recently, Chakraborty et al. [40] demonstrated a high concordance of three transcriptome-based gene metrics, including 76 gene signature (76GS), Kolmogorov–Smirnov test, and multinomial logistic regression, in quantifying the EMT score of a given cell line or clinical sample. To assess whether the CAF index is also superior to these novel EMT matrices, the 76 GS method was chosen and applied to the aforementioned EMT score computation and survival analysis [41]. Briefly, this method calculates the mean-centered weighted sum of 76 gene expression levels for each sample in a given cohort. Since the weight factor is the correlation coefficient with CDH1 expression, positive and negative scores indicate epithelial and mesenchymal phenotypes, respectively. As summarized in Figure A1, EMT scores derived from Salt et al. [12] (EMT_Salt) and Byers et al. [41] (EMT_76GS) reversely correlate with each other in both cell lines (Figure A1a) and clinical samples (Figure A1b,c). In univariate Cox proportional hazard model assessments, EMT_76GS was superior to EMT_Salt in the NCKU-OrCA-40T cohort (Figure A1b) but was inferior to EMT_Salt in the TCGA pan-cancer cohort (Figure A1c). Whereas in multivariate Cox proportional hazards models, the CAF index remains the only covariate that is with statistical significance, reinforcing the notion that the CAF index is a useful complement to the EMT score in prognosis prediction.

## 3. Discussion

By comparative transcriptomic analysis of NOG xenografts derived from OSCC cells with high and low EMT scores, the present study identified a four-gene signature consisting of TGFB2, TGFBR2, TGFBI, and FN1, as the CAF index. Meta-analysis of a betel-quid OSCC dataset and the TCGA pan-cancer cohort indicated that a high CAF index is statistically associated with poor overall survival. As CAF emerged as a critical factor for treatment failures, our results might provide new thoughts for curative medicine in clinical oncology.

In the seminal “consensus molecular subtyping of colorectal cancer” study [5], the consensus molecular subtype (CMS) 4 was noted with mesenchymal, stromal infiltration, TGFβ activation, angiogenesis, and worse relapse-free and overall survival; CMS1 was depicted as MSI/CIMP high, immune infiltration and activation, and worse survival after relapse. Despite the fact that NOG mice only preserve partial immune systems, these biological features are reminiscent of the OC3 xenografts disclosed here, including stronger cellular immune responses (Figure 2 and Appendix A) and higher stroma contents (Figure 3 and Appendix A). Indeed, compared to TW2.6, OC3 is non-tumorigenic in nude and NOD/SCID mice (personal communication), supporting the notion that OC3 is immunogenic in vivo. Molecularly, increased expression of genes involved in the MHC-I mediated antigen presentation (e.g., B2M, HLA-A, B, C, E), interferon signaling pathways (e.g., IFITs, IFI44/IFI44L, IFNARs, IFNGRs, CXCL9-11, STAT1), and apoptosis program (e.g., CASP8, TRAIL) were evidently detected in the OC3 tumor parts. Intriguingly, the super-enhancer of CD47, also known as the “don’t eat me signal” [31,42], was likely switched on during OC3 tumor formation, since a significant difference of CD47 expression between the OC3 and TW2.6 tumors was noticed (Appendix A). Taken together, our results suggest that in vivo OC3 appeared to behave like a CMS1 and CMS4 hybrid in colorectal cancer. Whether immune surveillance and/or immunoediting were a cause for later stromal infiltration awaits further investigation.

The most prominent features in the OC3 stroma are Fn1 and ligands/receptors of the Tgfβ signaling pathway (Figure 3a). Through querying of the expression data of the OSCC single-cell RNA-seq project, the cell source present in the OC3 stroma responsible for expressing Fn1 and Tgfbr2 was confined to fibroblasts (Figure 4). Further, to correlate the xenograft findings with clinical relevance, the Cox proportional hazards model was employed. We anticipated that genes involved in CAFs infiltration of OSCC tissues should have a hazard ratio > 1. Significantly, other than TGFB1, TGFB3, and TGFBR3, the hazard ratios of the other five TGFβ axis genes and FN1 all fulfilled our criteria in which TGFB2 and TGFBI achieved statistical significance (Appendix A). Thus, we propose that the summed expression of TGFB2, TGFBR2, TGFBI, and FN1 might be an index for CAFs infiltrated in a bulk tissue.

CAFs as a whole are usually considered as resistance factors in cancer therapy. In pancreatic ductal adenocarcinoma, Evans’ lab first illuminated the use of a vitamin D receptor ligand (calcipotriol) to alleviate fibrosis in pancreatitis and human tumor stroma [43]. In colorectal cancer, treatment-naive tumors with high CAFs had a significantly shorter disease-free survival, and resistance to radiotherapy when accompanied by additional stromal signatures [44]. In the phase I clinical trial EDALIN, 3 of 12 patients with metastatic triple-negative breast cancer who received combined chemotherapies of stroma (sonidegib) and tumor (docetaxel) showed clinical benefits, including one patient experiencing a complete response [45]. Recently, in five head and neck patient-derived xenograft mouse models, Yegodayev et al. [18] demonstrated that TGFβ-activated stromal CAFs limited cetuximab efficacy in vivo, which can be improved by an SMAD3 inhibitor, SIS3. Given these promising results, comprehensive consensuses about the roles of CAFs in tumorigenesis are emerging [15,46].

Of note, FN1 is the only molecule overlapping in the gene matrices of the EMT score and CAF index (Figure 5a), implying its essential role in tumor progression. FN1 is a class of adhesive glycoproteins that participate in ECM functions of neoplastic cells, including cell adhesion, proliferation, and migration. With regard to TME ECM, FN1 is essential for initial incorporation and maintenance of latent TGFβ binding protein 1 (LTBP1) and TGFBI [47,48]. FN1 assembled in the tumor stroma initiates a cascade of cellular signaling pathways beneficial to cancer progression [49]. Of particular importance, preclinical and clinical trials using FN1-guided cancer therapies, e.g., the extra domain A and B (EDA/B) of FN1 variants specific to tumor, are areas of active investigation [21,48].

## 4. Materials and Methods

### 4.1. Cell Culture

CGHNC9 [50], OC3 [51], OEC-M1 [52], and TW2.6 [53] were kindly provided by researchers at distinct institutions in Taiwan. All cells were cultivated in specified media as described in the original literature or following the manufacturer’s instructions. Cells were grown in a humidified 37 °C incubator with 5% CO_2_.

### 4.2. Antibodies

The antibodies used in this study were purchased from the following sources: anti-KRT18 (#4548), anti-STAT1 (#9172), and anti-TNFSF10 (#3219) were from Cell Signaling (Danvers, MA, USA); anti-Fn1 (AB2033) was from Millipore (Billerica, MA, USA).

### 4.3. Immunohistochemical Staining

Sections were dewaxed, rehydrated, and incubated with Trilogy^TM^ (Cell Marque, Rocklin, CA, USA) at 121 °C for 10 min to unmask antigens. At room temperature (RT), the slides were immersed in 3% hydrogen peroxide for 15 min to quench endogenous peroxidase activity; and then 1% bovine serum albumin for 60 min to block nonspecific antigenic sites. Slides for single staining were incubated with indicated primary antibodies at 4 °C overnight. After washing with TBS, slides were incubated with horseradish peroxidase (HRP)-conjugated secondary antibodies (#K5007, DAKO, Glostrup, Denmark) for 30 min at RT. Slides for dual immunohistochemical staining (Figure 3a and Appendix A) were incubated with anti-Fn1 and anti-KRT18 at 4 °C overnight. After washing with TBS, slides were incubated with horseradish peroxidase (HRP) and alkaline phosphatase (AP)-conjugated secondary antibodies (MRCT525, Biocare Medical, Pacheco, CA, USA) for 30 min at RT. Color development of the tissue section was conducted by using chromogens diaminobenzidine (Agilent, Santa Clara, CA, USA) for HRP and StayGreen (Abcam, Cambridge, UK) for AP. All slides were counterstained with Mayer’s hematoxylin. Slides were scanned by a Pannoramic MIDI scanner (3DHISTECH, Budapest, Hungary). To quantitate the immunostaining of KRT18 and Fn1 in dual IHC (Figure 3) and collagen fibers in Masson’s trichrome stain (Appendix A), scanned images at 100× magnification were digitized and quantitated by using Immunohistochemistry (IHC) Image Analysis Toolbox in ImageJ (NIH, Bethesda, MA, USA) according to the software documentation.

### 4.4. Animal Experiment

For each mouse, half a million mycoplasma-free TW2.6 or OC3 cells mixed with an equal volume of Corning^®^ Matrigel^®^ matrix (Corning, NY, USA) were subcutaneously implanted into the flank of 10–16- (Exp1) or 11 (Exp2)-week-old NOG. We measured the tumor size and mouse weight twice a week. For Exp1, all tumors were collected on day 68. For Exp2, to obtain tumors with ~500 mm^3^ in size, tumors of TW2.6-NOG were collected on day 32; tumors of OC3-NOG were collected on day 81. All procedures were approved by the Institutional Animal Care and Use Committee of National Health Research Institutes, Taiwan (Protocol No: NHRI-IACUC-106057-M1-A. Approval date: 1 February 2018).

### 4.5. mRNA-Seq Analysis

Total RNA from xenograft tumor tissue was homogenized using MagNA Lyser Green beads (Roche, Indianapolis, IN, USA) by MagNA Lyser (Roche, Indianapolis, IN, USA) followed by extraction using TRIzol^®^ reagent (Invitrogen Life Technologies, Carlsbad, CA, USA) and cleaned up using the RNeasy kit (Qiagen, Hilden, Germany) according to the manufacturer’s instructions, including the optional DNase I digestion step. The purity and concentration of total RNA was measured using a Thermo Scientific NanoDrop 2000 UV-Vis Spectrophotometer (Thermo Scientific, Wilmington, DE, USA). Total RNA integrity was assessed by an Agilent Bioanalyzer and the RNA Integrity Number (RIN) was calculated; samples that had an RIN > 7 were selected for RNA amplification and sequencing. mRNA-seq experiments for OC3-NOG08 and TW2.6-NOG14 were carried out by the VYM Genome Research Center of National Yang-Ming University, Taiwan. All the other mRNA-seq experiments were performed at the Institute of Molecular Biology (IMB) Genomic Core Lab of Academia Sinica, Taiwan.

### 4.6. Bioinformatics Analysis

OSCC cell line cohort: The normalized RNA-seq dataset was downloaded from the NCBI gene expression omnibus (GEO, https://www.ncbi.nlm.nih.gov/gds) under accession number GSE150469. The EMT score for each cell was calculated by summation of the 9 mesenchymal genes followed by subtraction of the sum of 5 epithelial genes (Figure 1a). The Log2 value was applied.

Xenograft datasets: To identify differentially expressed genes (DEGs) and biological pathways enriched in the human (tumor) and mouse (stroma) reads, normalized human and mouse expression matrices were subjected to *GSEA* [26] (Appendix A) and R package *limma* [27] analyses. Selected DEGs (FC > 1.5) were further subjected to gene ontology and pathway enrichment analysis by *DAVID* [28] (Figure 2a and Figure 3a). Differences were considered significant when *p* < 0.05.

OSCC scRNA-seq, NCKU-OrCA-40TN, TCGA pan-cancer, and MET500: The raw expression matrix of the single-cell OSCC RNA-seq dataset was downloaded from UCSC Cell Browser (https://cells.ucsc.edu/). Expression values (TP100K) were normalized, scaled, and log-transformed by using R *Seurat* (v.4.0.0) [54]. *DotPlot* was used to visualize the expression of FN1 and TGFβ axis genes across nine cell types comprising 5902 cells (Figure 4b). The normalized microarray dataset of NCKU-OrCA-40TN [37] was downloaded from NCBI GEO (GSE37991) (Figure 5). Patient information and gene expression for the 9356 patients in GDC TCGA (PANCAN) and of 811 met samples in MET500 were obtained through the UCSC Xena portal [39] (Figure 6). Wherever applicable, the EMT score and CAF index were calculated by using Log2 values. Kaplan–Meier survival analysis and Cox proportional hazards model were performed by using R *survival*, *coxph*, and *ggforest*. The Pearson correlation test of the CAF index and tumor content was evaluated by *cor.test*. In Figure 1c, Figure 3c, and Appendix A, the unpaired two-tailed *t*-test was applied to evaluate the statistical differences by GraphPad Prism 5.0. Code availability: https://github.com/sflinlab/2020-paper-CAF.idx

## 5. Conclusions

In the present study, we provided evidence that OSCC tumor cells with a high EMT score were associated with an elevated stromal content. Correlational analysis of clinical OSCC datasets with TGFβ axis differentially expressed genes revealed that the summed expression of FN1, TGFB2, TGFBR2, and TGFBI, dubbed as the CAF index, is a powerful predictor in dichotomized survival analysis for patients of diverse cancer subtypes. As the oncogenic roles of otherwise non-malignant CAFs are beginning to emerge, our results might illuminate new targets for curative therapies in clinical oncology.

## Figures and Tables

**Figure 1 cancers-12-01718-f001:**
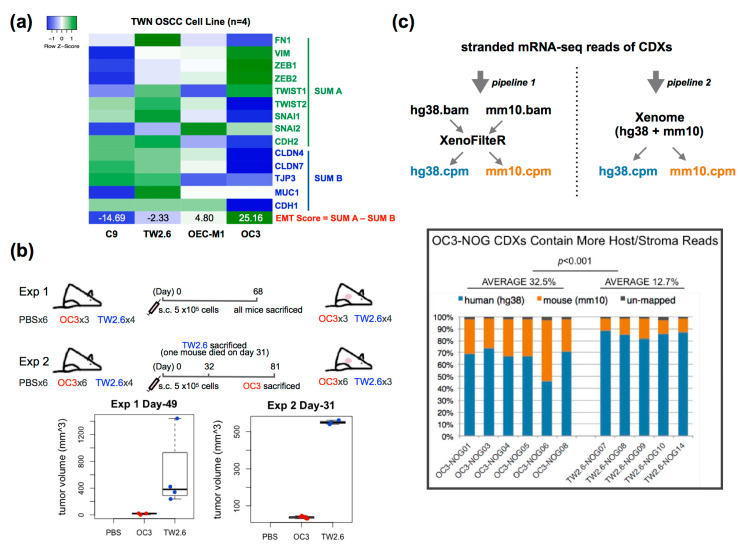
The oral squamous cell carcinoma (OSCC) cell line with a high epithelial–mesenchymal transition (EMT) score is associated with strong stromal infiltration in vivo. (**a**) Heatmap showing expression of 14 EMT genes [12] and derived EMT scores in four oral cancer cell lines: OC3, C9, OEC-M1, and TW2.6; (**b**) Schematic workflow of animal study. s.c., subcutaneous injection. Box plots show smaller tumor volumes were detected in OC3-NOG mice compared to that in the TW2.6-NOG mice in two independent experiments; (**c**) RNA-seq analysis of cell line-derived xenografts (CDXs) from OC3- and TW2.6-NOG mice. (Upper) Reads of tumor cells (hg38) and stromal constituents (mm10) for each CDX were determined by using two pipelines published previously [24,25]. Results from both methodologies revealed OC3 CDXs contain higher fractions of stroma reads. Bar plot shows percentages of tumor and stroma reads in each CDX. Student *t*-test was used to evaluate the difference between OC3 and TW2.6 mouse reads derived from pipeline 1.

**Figure 2 cancers-12-01718-f002:**
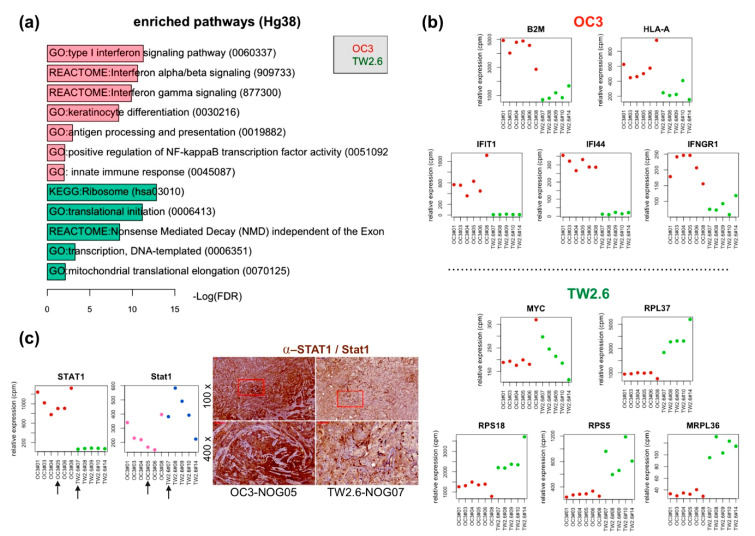
OC3 tumor cells displayed a stronger innate immune response in xenograft tissues. (**a**) Enriched gene ontology terms (fold change > 1.5, FDR < 0.05) in OC3 tumor cells (red) and TW2.6 tumor cells (green); (**b**) Plots of indicated differentially expressed genes in OC3 (upper) and TW2.6 (lower) tumor cells. (**c**) (Left) Plots denote RNA-seq reads of human STAT1 and murine STAT1 in OC3 and TW2.6 CDXs. Arrows indicate samples used for immunohistochemical (IHC) staining. (Right) Representative IHC micrograph images of indicated xenograft sections stained for human and murine STAT1 (brown). Scale bar, 50 μm.

**Figure 3 cancers-12-01718-f003:**
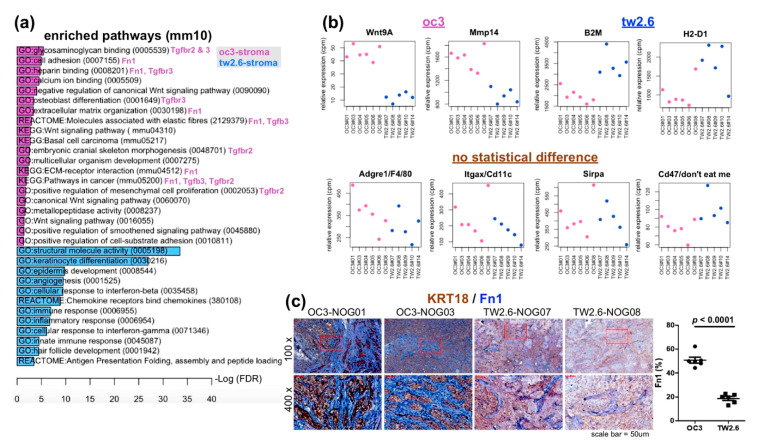
OC3 stromal components displayed upregulated expression of Fn1 and TGFβ-related cellular pathways. (**a**) Enriched gene ontology terms (fold change > 1.5, FDR < 0.05) in OC3 (orchid) and TW2.6 stroma cells (royal blue); (**b**) Plots of indicated differentially expressed genes in the OC3 (oc3) and TW2.6 stroma (tw2.6). Expression levels of genes indicative of macrophages (Adgre1, Sirpa), dendritic cells (Itgax), and cells responsive to immune surveillance (Cd47) in two groups of xenografts are also depicted (no statistical difference); (**c**) Representative immunohistochemistry images of indicative sections co-stained with human KRT18 (brown) and murine Fn1 (blue). Quantitative comparisons of all samples (also refer to Appendix A) are summarized on the right.

**Figure 4 cancers-12-01718-f004:**
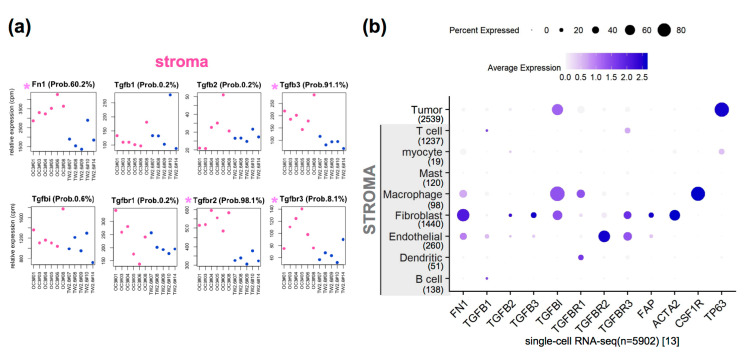
Associations of marker genes to fibroblast infiltration in OC3 xenograft tissues. (**a**) Plots denote expression levels of the murine Fn1 and TGFβ axis genes in each OC3 and TW2.6 CDXs. Probability (Prob.) refers to the *limma* [27] evaluated likelihood that the indicated gene is differentially expressed in the two tested groups. (**b**) Dot plot denotes RNA expression of indicated genes (*x*-axis) across nine cell types (*y*-axis) composing the single-cell OSCC RNA-seq dataset (GSE103322). Dot size indicates the proportion of cells within indicated cell type expressing indicated gene; color intensity represents binned count-based expression level (log(scaled normalized_count + 1)) amongst expressing cells. TP63 serves as a tumor cell marker.

**Figure 5 cancers-12-01718-f005:**
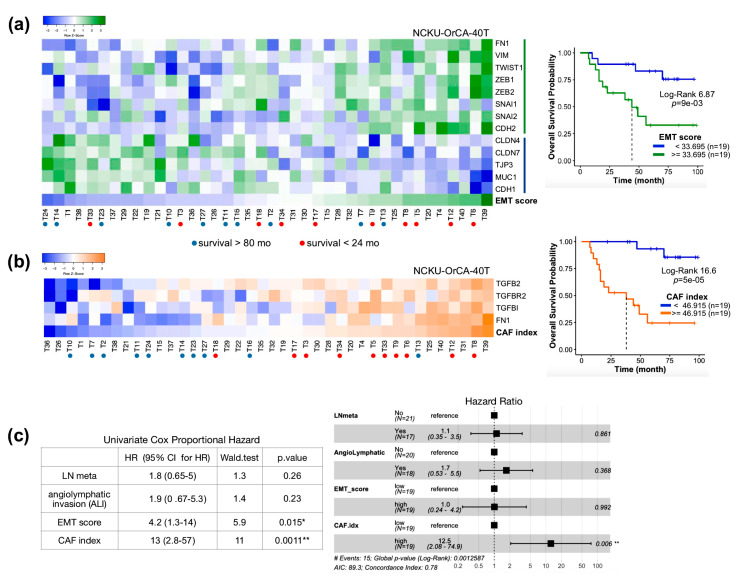
Prognosis evaluation of NCKU-OrCA-40TN (GSE37991) by using the EMT score and CAF index. (**a**) Heat map showing the expression of 14 EMT genes [12] and the EMT score for each tumor tissue (TWIST2 is not available in this dataset). Kaplan–Meier survival plot and Log-rank test statistics are denoted on the right. (**b**) Heat map showing the expression of TGFB2, TGFBR2, TGFBI, FN1, and the summation of the four genes (CAF index) in the 40 tumor tissues. Kaplan–Meier survival plot and Log-rank test statistics are denoted on the right. Patients with top 10 longest and shortest survival times are denoted with blue and red dots, respectively. Note T28 and T32 were not included due to treatment-related death. (**c**) Univariate (left) and multivariate (right) Cox model assessments of lymph node metastasis (LN meta), angiolymphatic invasion, EMT score, and CAF index. * *p* < 0.05, ** *p* < 0.01.

**Figure 6 cancers-12-01718-f006:**
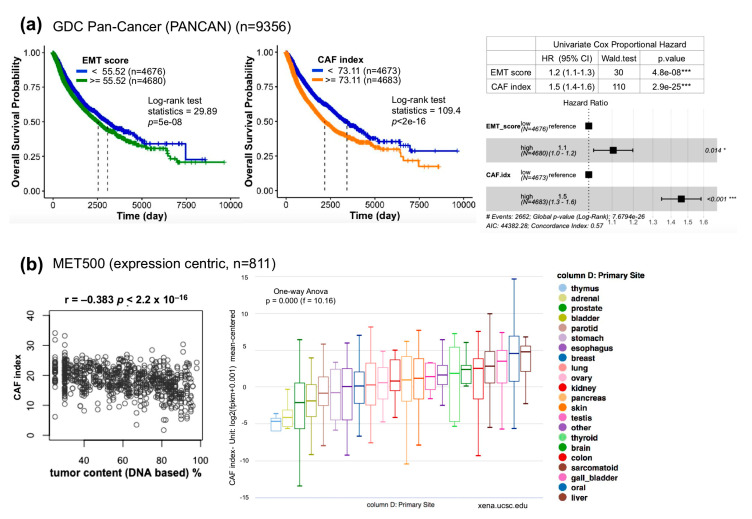
Clinical relevance of the CAF index. (**a**) Kaplan–Meier survival plots of GDC TCGA pan-cancer dataset by using the EMT score (left) or CAF index (right). Univariate and multivariate forest plots of Cox proportional hazards model assessments are summarized to the right. (**b**) Index of CAFs in metastatic cancers [38]. (Left) Pearson correlation analysis between the CAF index and tumor content estimated from exome DNA-seq using CNVs and SNVs. (Right) Box plot denotes CAF indices of each primary cancer type in the MET500. Data access and meta-analyses were performed online through the UCSC Xena portal [39]. Blank areas of the box plot screenshot were cropped for the interest of better visualization. * *p* < 0.05, *** *p* < 0.001.

## Data Availability

The raw and processed xenograft data generated in the current study are available through the Gene Expression Omnibus (GEO, https://www.ncbi.nlm.nih.gov/geo/) with accession number GSE149496. The raw and processed Taiwanese oral cancer cell line data are available through GSE150469.

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
