# Peer review of "Index of Cancer-Associated Fibroblasts Is Superior to the Epithelial–Mesenchymal Transition Score in Prognosis Prediction"

_cancers, 2020, doi:10.3390/cancers12071718_

Round 1

Reviewer 1 Report

In this research authors identify a 4-gene index (FN1, TGFB2, TGFBR2 and TGFBI), denoted as iCAFs, which was shown to have prognostic significance in OSCC and other cancer subtypes. In this study authors clearly describe the steps taken that led to the development of iCAFs. I find the study interesting, novel and well performed. The abstract however should have the abbreviations (OSCC, TCGA) clarified when first used.

I recommend this manuscript for publication.

Author Response

Thank you very much for your kind comments!

Just want to relay to you that we updated Figure 4b in which a dot plot was used to illustrate multiple gene expression across 9 cell types in the single-cell OSCC RNA-seq project. We feel dot plot is easier to understand and more informative. None of our conclusions was affected by this replacement.

Also that another reviewer reminded us that iCAFs is easily confused with inflammatory CAFs, which has been used by others in the field. Thus we changed ours to CAF index in the revised MS.

Please find revised Figure 4 in the attached pdf.

Reviewer 2 Report

The authors present an interesting study introducing a new score for evaluation EMT transition in different types of cancer. The study further highlights the importance of CAFs in the pathogenesis of tumours. However, there are some concerns regarding the EMT score being used.

Major Comments

  • The concept of EMT score is not introduced properly in the introduction. Needs to be clarified.
  • The authors do not explain why the Salt et al. EMT score has been used particularly in the study. As there are many other EMT scores available (76-gene EMT signature, BATTLE trial (Byers et al., 2013), Kolmogorov–Smirnov test  Tan et al., 2014, multinomial logistic regression MLR George et al., 2017) to name a few.  Can the authors compare these EMT scores in their study as well to make the finding more concrete?
  • Only one cell line from each category (high and low EMT score) is used throughout this study. This is a major weakness of the study. At least two cell lines per category should be considered a minimum according to my opinion.
  • Please change the abbreviation for iCAFs to something else, as this can be confused with the established term inflammatory CAF that is abbreviated iCAF.

Minor comments

  • No explanation to EMT score and CAF index is provided in the abstract, which makes it difficult to understand what the study is about just by reading the abstract. The text needs to be work on to make the abstract more understandable
  • Abbreviation for OSCC and TCGA are missing in the abstract (Page 1, Lines 34 and 38)
  • Why was TW2.6 used as cell line with low EMT score and not C9? Please explain on what basis the decision to use TW2.6 in this study was taken.
  • How can a cell line with high EMT score give rice to smaller tumours than a cell line with low EMT score (fig. 1b)? This contradict the initial statement that EMT is associated with poor prognosis. Please discuss around this.
  • Please remove the abbreviation for DEG on (page 4, line 135) as it has already been opened earlier (page 4 line 130)
  • Fig 2b, first row, the labels of the x-axis has been cut.
  • On line 194-195 masson’s richrome stain is mentioned. Where is the figure supporting this?

Author Response

Please see the attachment, thanks!

Reviewer 3 Report

Ko YC et al have analysed by RNA Seq human OSCC cell lines xenografts, studying both the expression profile of the tumors and the microenvironment. From this analysis, the authors describe a new index termed iCAF, that uses the expression levels of FN1 and TGFb expression, in order to predict tumour prognosis, not only in OSCCs tumors, but also in other cancer types.  This is a very interesting study, showing how few genes expressed in the tumour stroma can be used as a prognosis tool in different cancer types.

The authors observe that the OC3 tumors have a strom enriched in ECM and TGF-beta expression, while the tumour cells are enriched in genes related to inflammatory response. It would be interesting to see if those genes also correlate with tumour survival, since the authors propose that the immune surveillance response might be the cause of stromal infiltration.

It would also be informative, if the authors could also address if the iCAF could be used as a prediction of response, if there are available datasets of OSCC patients with different treatments.

Minors:

  • In the Abstract, the authors should define what’s OSCC, line 34, when it’s first mentioned.
  • Line 213, precious should be previous
  • It would be very useful if authors made the code they have used available.

Author Response

Point 1: The authors observe that the OC3 tumors have a stroma enriched in ECM and TGF-beta expression, while the tumour cells are enriched in genes related to inflammatory response. It would be interesting to see if those genes also correlate with tumour survival, since the authors propose that the immune surveillance response might be the cause of stroma infiltration.

Response 1: Yes, indeed we were very curious about the spatiotemporal interactions between tumor cells, immune system, and CAFs. In the same study, TW2.6 was extremely immune-evasive. While the murine/host innate immune cells were similar between the oc3- and tw2.6 stroma enumerated by CIBERSORT, we detected significantly high tumor-associated macrophage (TAM) scores (Pollard Lab, PMID 30930117) in the tw2.6-stroma. I personally believe that the original OC3 cells in their patient body must have had evolved a way to escape the immune attack, just like they survived to grow in the NOG mice we used here. I would love to assess your question as soon as HSC-humanized NOG mice are available, then I can obtain a complete set of immune response genes in a more physiologically relevant condition :-))

Point 2: It would also be informative, if the authors could also address if the iCAF could be used as a prediction of response, if there are available datasets of OSCC patients with different treatments.

Response 2: We are prepare IRBs to obtain more clinical informations as well as more samples to address this important question! Thank you very much for your comment!

Minor: It would be very useful if authors made the code they have used available. 

Response 3: Thanks for your encouragement. I understand this is crucial to "Reproducible Research". So the link to my ugly codes is noted on p13, line 457 in the revised manuscript. 

_______

Additional note we want to relay here is that we updated Figure 4b in which a dot plot was used to illustrate multiple gene expression across 9 cell types in the single-cell OSCC RNA-seq project. We feel dot plot is easier to understand and more informative. None of our conclusions was affected by this replacement.

Also note that another reviewer reminded us that iCAFs is easily confused with inflammatory CAFs, which has been used by others in the field. Thus we changed ours to CAF index in the entire MS.

Please find revised Figure 4 in the attached pdf for your reference.

Round 2

Reviewer 2 Report

The authors have really made an effort to respond to my previous comments. As I can see, all of my concerns in the first round of review has been addressed properly. I just have one minor comment. In the abstract it is stated that “By transcriptomic analysis of xenograft tissues, we found that oral squamous cell carcinoma (OSCC) tumor cells with a high EMT score are associated with…”. What do the authors mean with a high EMT score? As a reader, only reading the abstract, it is difficult to fully understand this, since the concept of EMT score hasn’t been introduced anywhere in the abstract.

Author Response

Dear reviewer,

Really appreciate all your input on our manuscript!

For the abstract, now I included a sentence after the first appearance of EMT score, run like this:

By transcriptomic analysis of xenograft tissues, we found that oral squamous cell carcinoma (OSCC) tumor cells with a high EMT score, the computed mesenchymal likelihood based on expression signature of canonical EMT markers, are associated with elevated stromal contents featured with Fn1 and Tgfβ-axis gene expression.

Hope this works !